# Penetration Gain Study of a Tungsten-Fiber/Zr-Based Metallic Glass Matrix Composite

Feng Zhou [1], Chengxin Du [1,2,*], Zhonghua Du [1], Guangfa Gao [1], Chun Cheng [3] and Xiaodong Wang [1]

1   School of Mechanical Engineering, Nanjing University of Science & Technology, Nanjing 210094, China; zhoufeng199191@163.com (F.Z.); duzhonghua@aliyun.com (Z.D.); gfgao@ustc.edu.cn (G.G.); 18752006367@163.com (X.W.)
2   School of Materials Science and Engineering, Nanjing University of Science & Technology, Nanjing 210094, China
3   Impact and Safety Engineering, Ningbo University, Ningbo 315211, China; xiangchun893@163.com
*   Correspondence: duchengxin4324@163.com

**Abstract:** A tungsten fiber/Zr-based bulk metallic glass matrix composite (Wf/Zr-MG) is a potential penetrator material. To compare and analyze the penetration behavior of Wf/Zr-MG and a tungsten heavy alloy (WHA), a penetration experiment into the 30CrMnMo homogeneous armor target plate (RHA) is conducted in the present paper, by using a Φ37 mm smooth bore artillery with an impact velocity of 1550 ± 40 m/s. Unlike the penetrator made of WHA, the self-sharpening phenomenon was observed in the nose of the Wf/Zr-MG rod. The experimental results indicate that the penetration ability of Wf/Zr-MG rod is approximately 10% higher than that of the WHA rod when the impact velocity is 1550 ± 40 m/s. The combined findings on the microscopic morphology, composition, hardness distribution around the crater, and the macroscopic structure of the penetrator residual show that under this impact velocity, the Wf/Zr-MG material shows amorphous gasification. The Wfs outside the rod shows bending and backflow, resulting in the maintenance of the self-sharpening nose of the penetrator during the penetration process. Moreover, the hardness peak around the crater formed by the Wf/Zr-MG rod is lower, and the penetration crater is straighter, indicating that the Wf/Zr-MG rod has a stronger slag removal ability, lower penetration resistance, and higher penetration efficiency. It is an ideal penetrator material.

**Keywords:** tungsten fiber/Zr-based bulk metallic glass matrix composite (Wf/Zr-MG); self-sharpening; microscopic morphology; high velocity penetration

## 1. Introduction

With the development of armor protection technology, more and more RHA materials with high hardness, high strength, penetration resistance, and avalanche resistance are widely used in the protection of typical targets, such as buildings, tanks, and ships. The protective ability of the system has significantly improved [1,2]. Higher requirements are put forward for the penetration capability of armor piercing projectile. Presently, the high-density WHA and depleted uranium alloy (DU) are the penetrator materials. The DU exhibits high shear sensitivity; it easily produces highly localized shear bands and presents the "self-sharpening" behavior in the process of penetration [3,4]. The WHA rod usually tends to assume a "mushroom" shape, which reduces its penetrating capability [5,6]. The penetrating performance of the DU rod is approximately 10–15% greater than that of WHA rod under the same conditions [7]. However, the DU rod is not accepted in the field of kinetic energy armor weapons [8], because it emits low radiation to humans and the environment. Therefore, a new type of penetrator material is urgently needed to improve the armor piercing performance. Zr based bulk metallic glass (Zr-MG) has good mechanical properties. For example, the quasi-static compression strength of Wf/Zr-MG is more than 1.8 GPa, and the dynamic compression strength of Zr-MG is more than

2.5 GPa [9,10], whereas the strength of WHA is only 1.2 GPa. At the same time, Zr-MG also has shear sensitivity; it may show a shear self-sharpening phenomenon if Zr-MG is used at a high penetration velocity similar to DU [11–15]. In addition, Zr-MG has a large supercooled liquid region, which can be used to prepare high quality bulk materials. However, the density of Zr-MG is only approximately 6.5 g/cm³. This kind of alloy has almost no macroscopic plasticity, which limits its application in the field of armor piercing engineering. Thus, metallic glass matrix composites, especially the Wf/Zr-MG, were recently developed. By adding Wfs to the Zr-MG, the Wf/Zr-MG can maintain high strength while showing a good adiabatic shearing performance and high plasticity [16,17]. At the same time, Wf/Zr-MG has penetration "self-sharpening" characteristics, and the density of Wf/Zr-MG increases to 17.1 g/cm³ when the volume fraction of Wfs is larger than 80%. Therefore, Wf/Zr-MG has become a focus of research in the field of armor piercing materials. Conner, Rong, Chen, Du, and Du conducted penetration tests involving the use of Wf/Zr-MG at a velocity of 1000–1700 m/s, and the penetration test results show that the penetration ability of Wf/Zr-MG results in an obvious "self-sharpening" phenomenon during penetration. Its penetration ability was better than that of WHA, and the average penetration depth was approximately 10% higher than that of the WHA [18–24]. The above research analyzed the penetration mechanism and failure mode of Wf/Zr-MG and WHA rod, but the analysis process was only reflected in the residual rod. The target is another factor that reveals the penetration gain mechanism of Wf/Zr-MG, for which the results have only been listed in relatively few articles, such as the literature [13–15]. Therefore, in order to enrich the research results in this direction, summarize and refine the previous research contents, and provide a new research idea for the research on the penetration mechanism of penetrators, it is particularly important to thoroughly explore the microstructure analysis of targets in the penetration process.

In the present work, we conducted penetration experiments with Wf/Zr-MG and WHA rods of the same size. The penetration capabilities of the two kinds of rods were compared. The microstructure around the crater penetrated by the two kind of rods on the target was analyzed by SEM, EDS, and a microhardness tester. The penetration gain of Wf/Zr-MG rod was revealed by the analysis results.

## 2. Experiment

### 2.1. Experiment Configuration

The rods were conducted by using a Φ37 mm smooth bore artillery. The experimental set-up is shown in Figure 1. The distance between the muzzle and the target was 16 m. The paper plates with on–off aluminum foil were used to measure the velocity of the rod and were arranged at a distance of 2 m away from the front of the target. The size of the target was 300 mm × 160 mm × 110 mm.

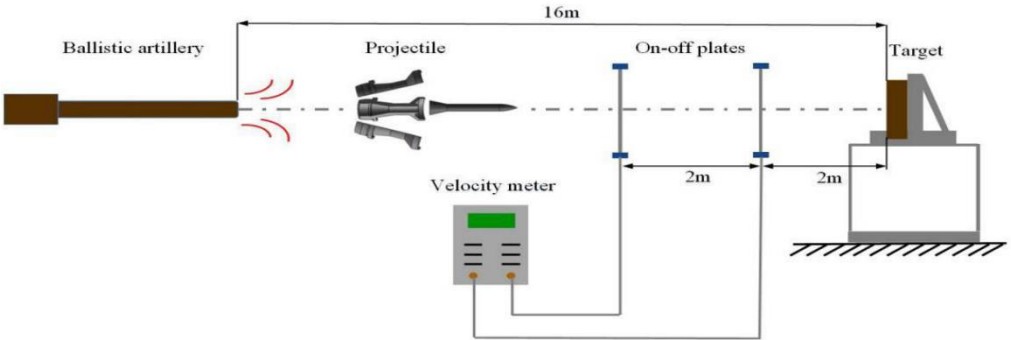

**Figure 1.** Experimental set-up.

### 2.2. Projectile Structure

The rod used in the experiment had a diameter of 10 mm, a length of 97 mm, a length-diameter ratio of 9.7, and a cone-like structure at the nose. The sabot for fixing the rod was

composed of 3 clips of 120°. The sabot separates from the rod when the projectile leaves the artillery. The specific structural parameters and the rod are shown in Figure 2.

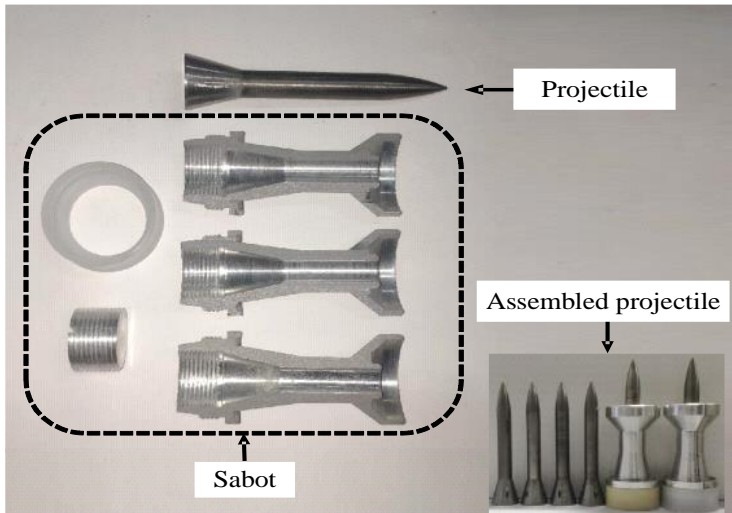

**Figure 2.** Projectiles used for impact experiment.

### 2.3. Projectile and Target Material

　　The WHA (WHA rods made of 93W) and Wf/Zr-MG were used in the experiment. In the Wf/Zr-MG rods, the Zr-MG (Zr41.25Ti13.75Ni10Cu12.5Be22.5) was the matrix, and the Wfs with a volume fraction of 80% and a diameter of 0.3 mm were uniformly distributed in a parallel manner in Zr-MG as the reinforcements. The density of Wf/Zr-MG was $17.1 \pm 0.05$ g/cm$^3$. The microstructure of Wf/Zr-MG is shown in Figure 3. Figure 3a shows that Wfs were uniformly distributed in a close packed state, and the solution is fully infiltrated without obvious pores. No obvious gap and interface reaction is found between Wf and Zr-MG, as shown in Figure 3b. Figure 3c shows the X-ray diffraction pattern of the composite. The body centered cubic (bcc) diffraction peaks for the Wf are superimposed on the broad diffuse scattering hump, which is typical characteristic from the amorphous structure. The BMG cross-sectional high resolution TEM pattern are shown in Figure 3d. Broad and diffuse diffraction rings are clearly seen in the TEM patterns, implying its amorphous structure, which is consistent with the XRD analysis results. The target used in the test was RHA with an average hardness of 320 HV. The mechanical properties of WHA, Wf, Zr-MG, and RHA are shown in Table 1.

**Table 1.** Main mechanical properties of all materials used in the experiment.

|  | WHA [25] | Wf [26] | Zr-MG [27] | RHA [28,29] |
|---|---|---|---|---|
| Density/(g·cm$^{-3}$) | 17.6 | 19.22 | 6.68 | 7.85 |
| Poisson's ratio | 0.3 | 0.28 | 0.36 | 0.3 |
| Modulus of elasticity/GPa | 360 | 370 | 96 | 210 |
| Yield stress/MPa | 1300 | 1725 | 1900 | 900 |

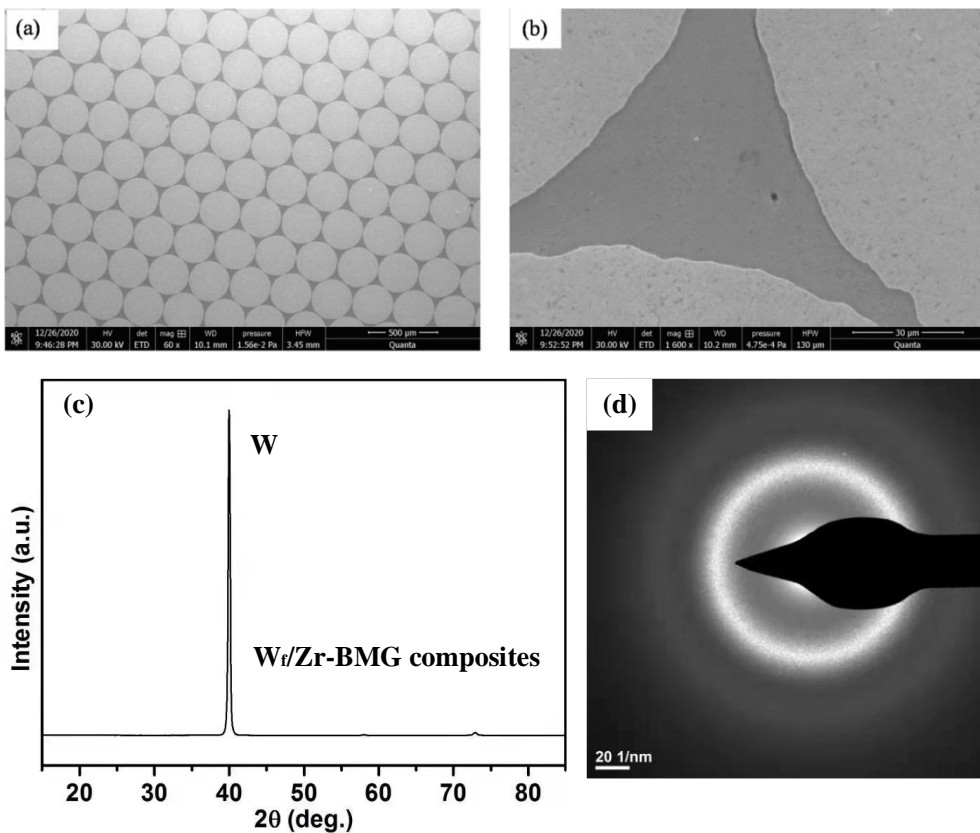

**Figure 3.** The microstructure characteristic of the Wf/Zr-based BMG composite, SEM micrograph in the back scatter mode of 80 vol.% Wf/Zr-MG composite (**a**); local zoom (**b**) (the light-colored section represents the tungsten fiber; the dark-colored part represents the Zr-based bulk metallic glass); X-ray of the composite (**c**); and TEM electron diffraction pattern of Zr-MG (**d**).

## 3. Penetration Test Result

A penetration experiment was carried out to compare the penetration performance of two kinds of rod materials on the RHA armor target. Three valid data were present in the experiment. Among these data, the mass, impact velocity, propellant mass, penetration depth, crater diameter, and the penetration depth ratio of the rods are shown in Table 2.

**Table 2.** Experimental data.

| Penetrator | | | Impact Velocity (m/s) | Target Plate | | | Penetration Depth Ratio (P/L) |
|---|---|---|---|---|---|---|---|
| Penetrator No. | Projectile Type | Mass of Rod (g) | | Penetration Depth (mm) | Average Diameter of Crater (mm) | Thickness of Target (mm) | |
| 1 | Tungsten alloy | 122.7 | 1585.8 | 95 | 18.7 | 110 | 0.98 |
| 2 | Wf/Zr-MG alloy A | 121.6 | 1553.1 | Perforated | 16.4 | 110 | 1.13 |
| 3 | Wf/Zr-MG alloy B | 121.9 | 1584.7 | Perforated | 17.1 | 110 | 1.13 |

### 3.1. WHA Rod

Figure 4 shows the longitudinal section of a crater penetrated by the WHA rods. The partial enlarged drawing of (a) shows that the surface of the crater penetrated by the WHA is uneven. There is a thin layer with a metallic luster on the surface of the crater. Many small granular structures are found locally, and these have obvious re-solidification characteristics. The partial enlarged drawing of (b) shows that the nose angle of the rod is approximately 114°, which is a clear "mushroom" shape.

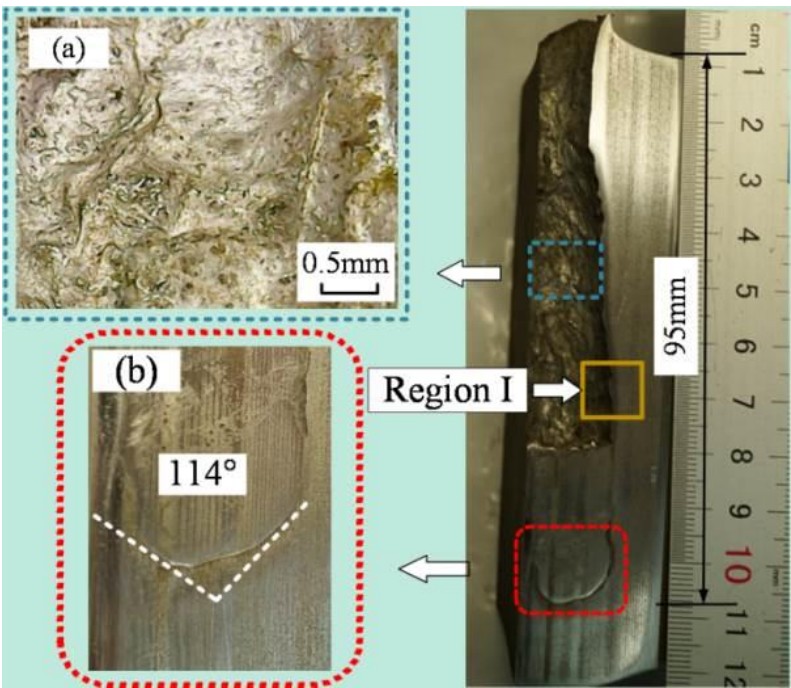

**Figure 4.** Longitudinal section of crater penetrated by the WHA rods. (**a**) partial enlarged dra- wing of craters; (**b**) head angles of the rods.

### 3.2. Wf/Zr-MG Rod

Figure 5 shows the longitudinal sections of craters penetrated by the Wf/Zr-MG rods. According to the partial enlarged drawing (a) in Figure 5A,B, the crater surface penetrated by the Wf/Zr-MG rod is obviously different from that penetrated by WHA, and there are many obvious "groove" marks scratched by Wfs on the crater surface. The partial enlarged drawing (b) in Figure 5A,B shows that the head angles of the rods are approximately 84° and 86°, respectively, showing a clear "cone" shape, and all of these angles are smaller than the nose angle of the penetrator residual of the WHA rod. Figure 5C shows a residual Wf/Zr-MG rod. The residual Wf/Zr-MG rod was porous. The outer part of the penetrator residual buckled and backflow was found along the axis. This phenomenon is similar to the research results of X.W. Chen [30], Z.H Du [23], and C.X. Du [24]. A part of Zr-MG gasification was due to a high temperature and strain rate during the high-speed penetration process, which caused the Wfs at the head and surface of the rod to lose restraint. Under high-speed penetration conditions, stress was higher than the buckling threshold of Wfs. The Wfs underwent dynamic buckling and backflow along the axis. The morphological features generated by scratching the surface of the crater are shown in an area (a) in Figure 5A,B.

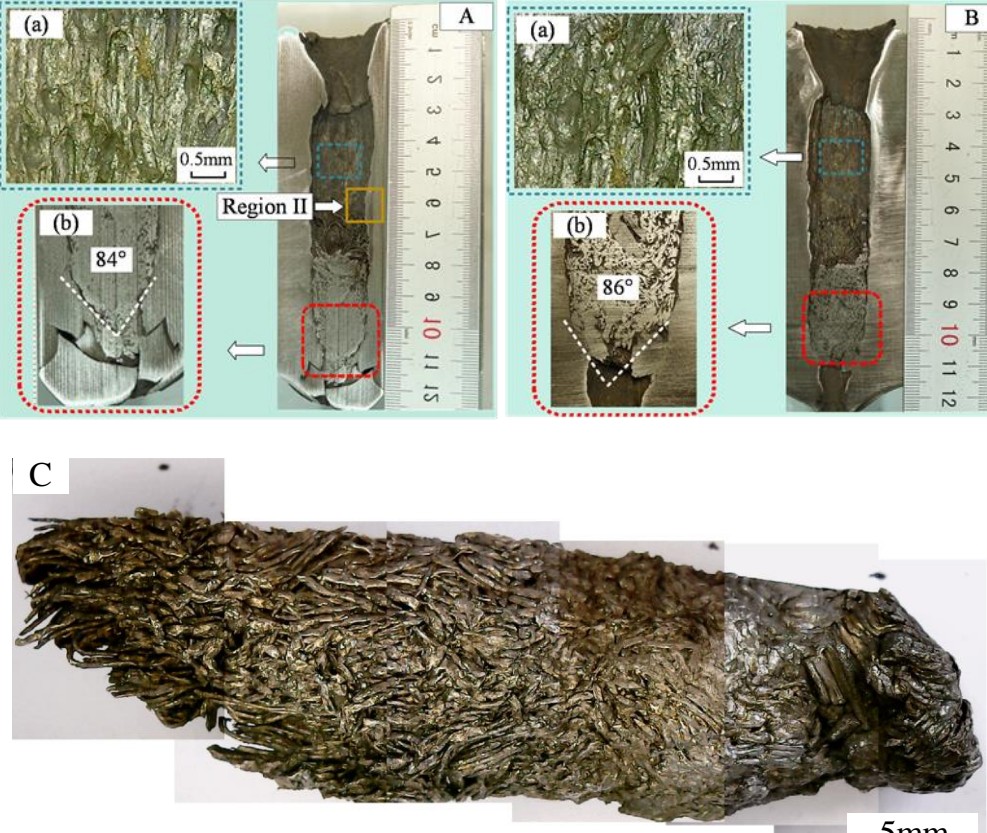

**Figure 5.** Longitudinal sections of craters penetrated by Wf/Zr-MG rods, (**A**) penetrator of 2#, (**B**) penetrator of 3#, and (**C**) residue of 2# rods; (a) partial enlarged drawing of craters; (b) head angles of the rods.

## 4. Micro-Structure Analyses of the Crater

### 4.1. Tungsten Alloy Penetrating Crater

Figures 6 and 7 shows the SEM image of Region I (Figure 4) and the EDS energy spectrum analysis of the crater penetrated by 93W rods. Figure 6 shows a light-colored uneven layer attached to the edge of the crater. According to the literature [31], the light-colored uneven layer is a melted and rapidly solidified layer (MRSL). The MRSL was closely bound to the target, and an obvious boundary existed between the MRSL and the target. The thickness of the MRSL was approximately 29 μm. During the penetration process of the WHA rod, the tungsten grains in the rods were severely compressed along the axial direction and flowed to the periphery, which led to the passivation of the nose of rods and the increase in the extrusion pressure of the rods on the wall of the crater. When the rods' plastic deformations become larger than the WHA plastic limit, the rods would break and peel off, which weakens the extrusion ability of the rods to the side wall of the crater. In the penetrating process of 93W rods, the above process was repeated many times, resulting in a wavy appearance at the interface between MRSL and target substrate, as shown in Figures 6a and 7a. In addition, the thickness of MRSL varied, indicating that in the process of penetration, the broken tungsten particles were passivated under the action of migration and drift of MRSL, which increased the backflow resistance of the material. Figure 6a shows that the MRSL included different materials, and the composition of the materials was determined by the EDS point scanning of metal fragments, as shown in Figures 6b–d and 7b. Among them, EDS 1–3 in Figure 6 reflects the element distribution of bright inclusions in MRSL, MRSL and matrix materials, respectively, and the EDS 1 in Figure 7b also reflects the element distribution of MRSL. The main composition in EDS 1 point was W, which proves that the bright part in the figure represents the broken tungsten

particles condensed in MRSL. It can be seen from Figure 6c,d that MRSL is similar to the matrix material, and its main component is Fe. Figures 6 and 7 show that the MRSL is composed of the melting target material and broken bulk tungsten particles.

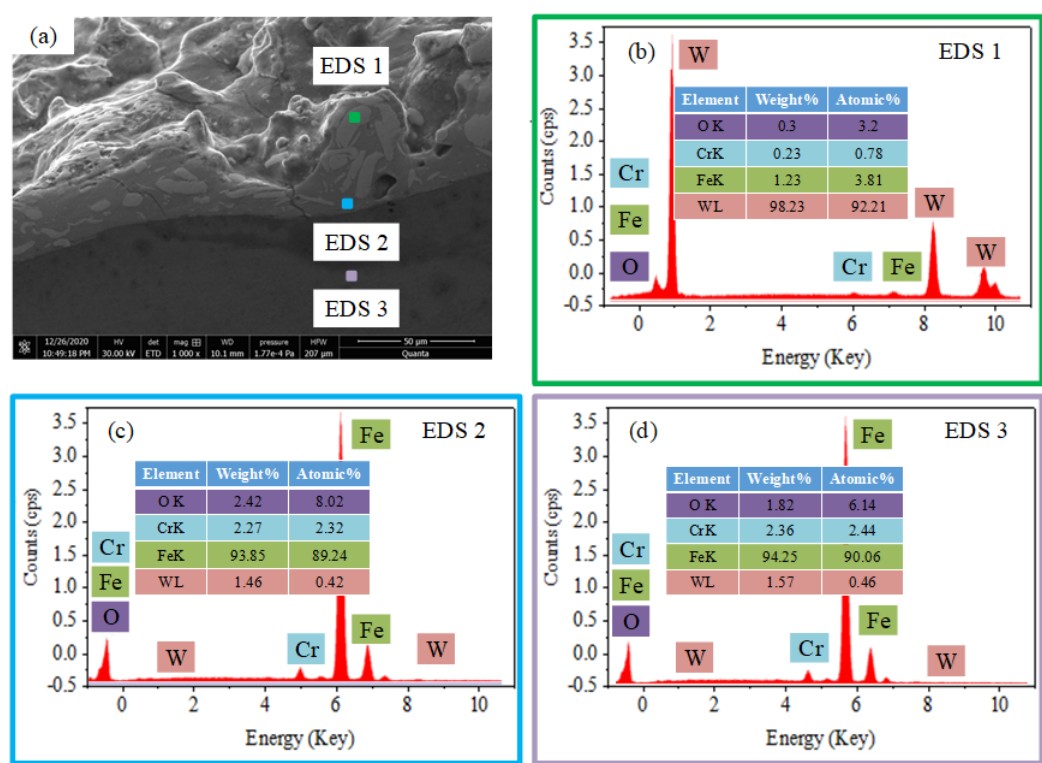

**Figure 6.** Section morphology and EDS spectra of the crater. (**a**) Region I of the 93W crater, (**b**) EDS 1, (**c**) EDS 2, and (**d**) EDS 3.

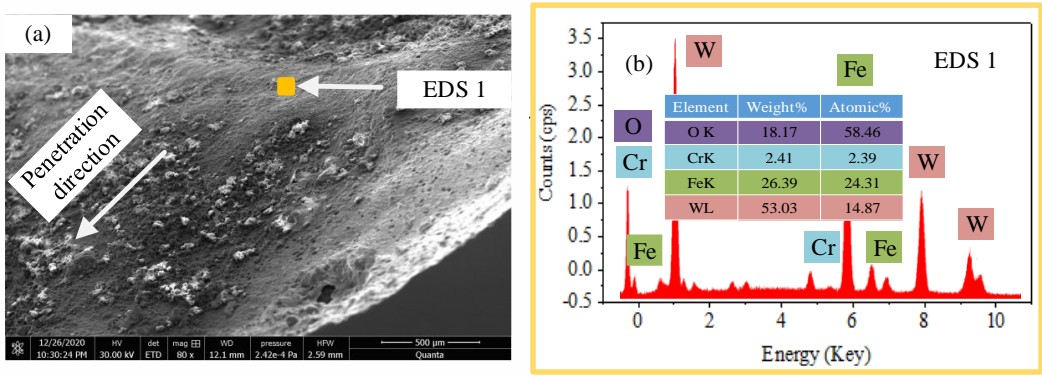

**Figure 7.** Internal surface morphology and EDS spectra of the crater. (**a**) Region I of the 93W crater and (**b**) EDS 1.

### 4.2. Wf/Zr-MG Alloy Penetrating Crater

Figures 8 and 9 show the SEM image of Region II (Figure 5) and EDS analysis of the crater penetrated by Wf/Zr-MG rods. Figure 8 shows that the MRSL is closely bonded with the RHA, and a clear demarcation is found between the MRSL and RHA. The thickness of MRSL formed by Wf/Zr-MG penetration was 18 µm. Unlike the crater penetrated by the WHA rods, the wall of the crater penetrated by Wf/Zr-MG rods was flat and straight on the whole. The SEM image (Figure 8a) shows that fractured Wfs are sporadically distributed in MRSL. Figure 5 comprehensively shows that most of the turned Wfs did not break, but coated both sides of the rod. Therefore, it was difficult for the Wfs to bond closely with micron MRSL. Under the joint action of the Zr-based metal gasification and vibration of

high-speed impact between rod and target, Wfs quickly fell off, and backflow was observed. Only a small part of Wfs broke during the penetration. Some micron-sized Wfs fragments and melting state W were wrapped on the target with a slower reflux rate and congealed on the crater surface.

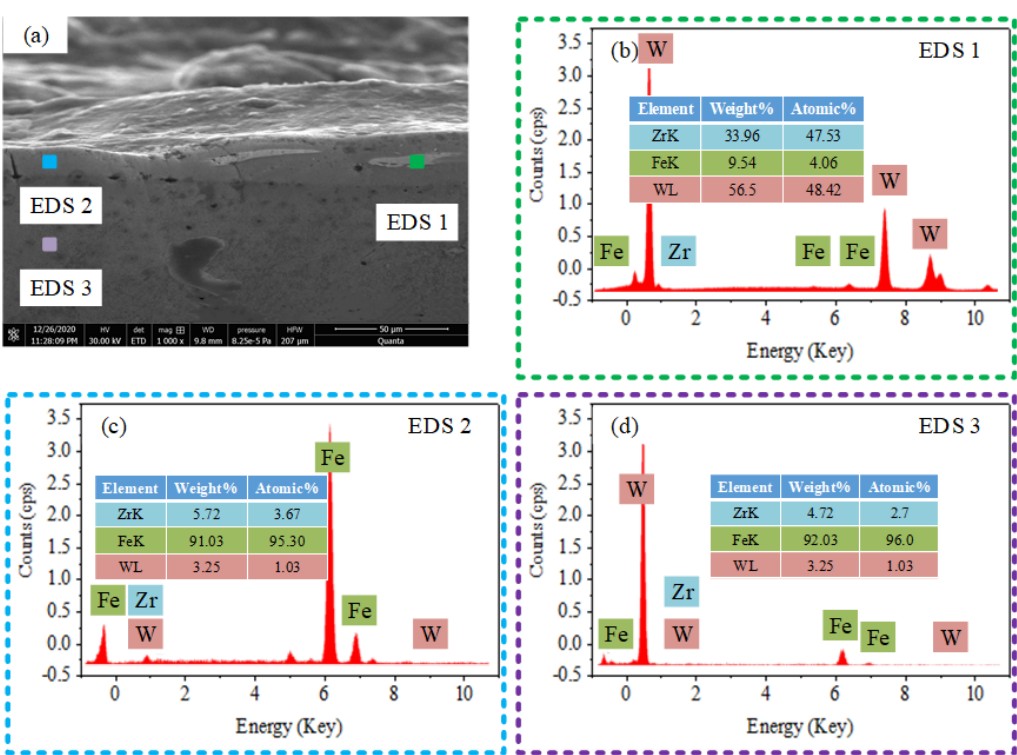

**Figure 8.** Section morphology and EDS spectra of the crater. (**a**) Region II of Wf/Zr-MG crater, (**b**) EDS 1, (**c**) EDS 2, and (**d**) EDS 3.

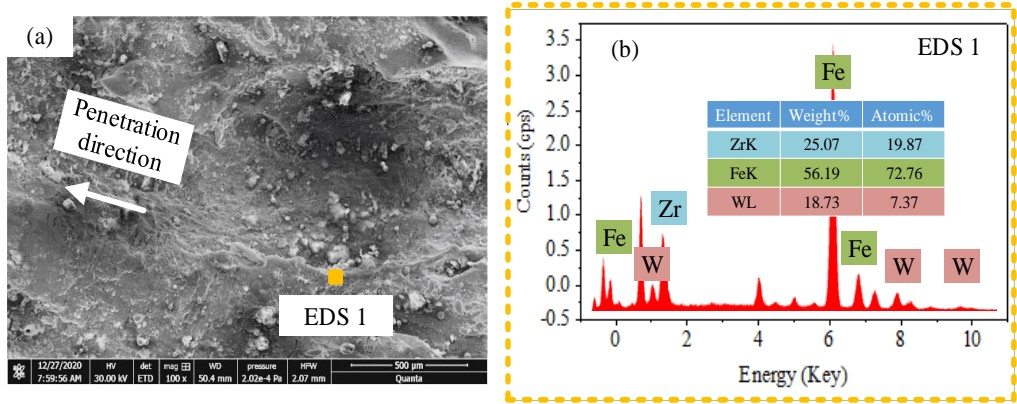

**Figure 9.** Internal surface morphology and EDS spectra of the crater. (**a**) Region II of Wf/Zr-MG crater and (**b**) EDS 1.

### 4.3. Target Hardness Distribution

The hardness distribution from the wall of the crater to the inside of the target plate can reflect the degree of plastic deformation of the target during the penetration. Therefore, the materials around the craters were cut from the target with a wire-electrode cutting machine and divided into a sampling size of 1.5 cm × 1 cm. The specific sampling situation is shown in Figure 10a,c. The HV1000 microhardness tester was used to test the hardness of the samples, and each sample was hit with four sets of hardness points. The interval between each set was 4 mm, and each set contained 10 hardness test points. The test points

extended inward along the side wall of the crater. The interval between each test point was 0.5 mm. The test load was 1 Kg, and the pressure was held for 10 s.

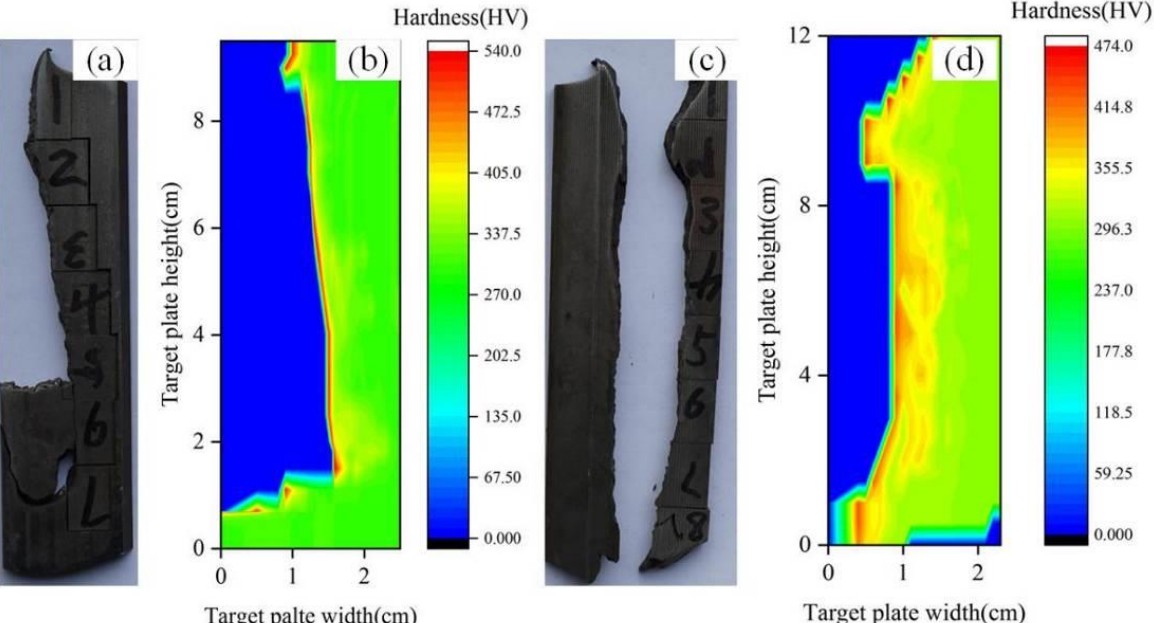

**Figure 10.** Target hardness distribution. (**a**) The 93W crater, (**b**) hardness distribution of the 93W crater, (**c**) Wf/Zr-MG crater, and (**d**) hardness distribution of the Wf/Zr-MG crater.

The hardness distribution of the wall of the craters penetrated by the two kinds of rods are shown in Figure 10b,d. The hardness distribution shows that work hardening occurs near the crater after the penetration. The hardness of the target decayed from the edge of the crater to the inside of the substrate, and the attenuation zoom was approximately 4 mm, which meant that the structure of the target was squeezed and deformed during the penetration, as shown in Figure 10b. The pressure of the WHA rod on the side wall of the crater is not linear, resulting in the hardness of the wall of the craters penetrated by the WHA rod not changing linearly (see Figure 10b,d, where there is a thin hardened layer of material that can be observed at the site of damage). It shows that the pressure of the projectile and MRSL improves the hardness of the crater's surface. Compared with the crater hardness penetrated by the WHA rod, the crater hardness penetrated by the Wf/Zr-MG rod (Figure 10d) presents an obvious gradient distribution. The gradient distribution indicates that the BMG releases heat and performs at a higher temperature, which is similar to the heat treatment on the target plate during the penetration process, and the higher temperature causes the hardness peak value of the side wall of the crater to reduce, and the hardness decreases in a gradient. The peak value of the crater's hardness penetrated by the 93W rod was 540 HV, whereas this was only 474 HV for the Wf/Zr-MG rod. Combined with the comprehensive analysis of the previous content, the self-sharpening effect of the Wf/Zr-MG rod improved the slag discharge ability of the crater and reduced the squeezing effect to the side wall of the target. Thus, the crater hardness penetrated by the Wf/Zr-MG rod was lower than that of the WHA rod. The reduction in the squeezing effect can reduce the penetration resistance, which was also one of the fundamental reasons for the stronger penetration ability of the Wf/Zr-MG rod.

## 5. Result and Discussion

### 5.1. Rod Erosion Mechanism of Two Materials

The above analysis results indicate that the erosion mechanism of WHA and Wf/Zr-MG differ. Figure 11 shows the process of the 93W rod's penetration into the RHA target. In the process of penetration, the compressive stress in the contact area between the projectile and target was more than the strength limit of the penetrator. The 93W rod showed good dynamic plasticity during the severe deformation. The nose shape significantly deformed, became blunter, flowed to the side, and became a clear "mushroom" shape. With increasing penetration depth, the severe plastic deformation continued to occur in the "mushroom" nose of the rod. The materials on the side of "mushroom" nose continued to erode and fell off from the rod. Tungsten fragments of different sizes were formed. The larger tungsten fragments were discharged from the crater under the reaction force of projectile impact, while the smaller tungsten fragments were coagulated on the surface of MRSL, and then the above process was repeated. The length of the rod gradually became shorter and, finally, the penetration process was completed.

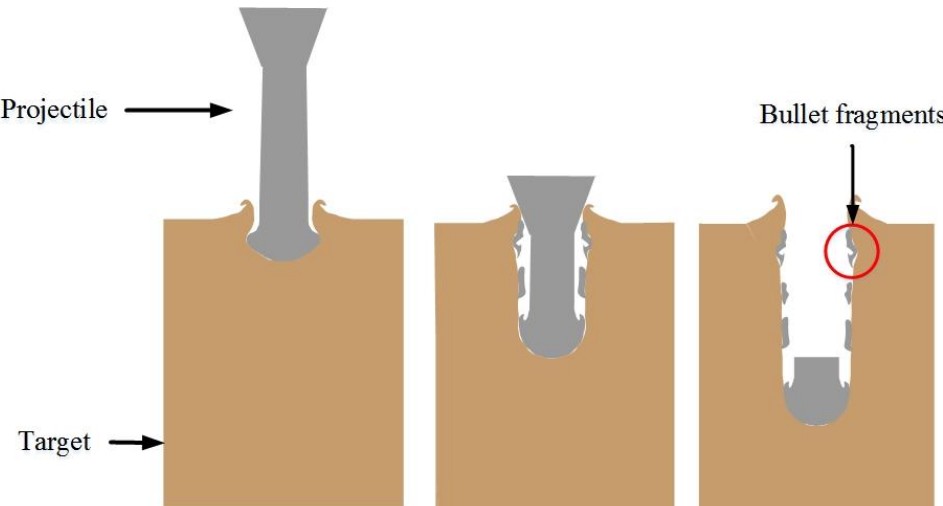

**Figure 11.** Penetration progress of RHA plates by the tungsten alloy projectile.

Figure 12 shows the process of the Wf/Zr-MG rod's penetration into the RHA target plate. As shown in the figure, the deformation of the initial rod nose is similar to that of the 93W rod, and the nose of the rod is blunt. The built-in Wfs were locally bent in a "broom" shape. With an increasing penetration depth, the thermal softening effect of the BMG attained the dominant position when the rod penetrated the target plate at a high speed [24], thereby greatly reducing the confinement ability of the embedded Wfs. Thus, the Wf/Zr-MG rod exhibited the dynamic buckling, fracture, and backflow phenomena of the Wfs when the Wf/Zr-MG rod penetrated the RHA target plate. With the increase in the penetration depth, the dynamic buckling, fracture, and backflow phenomena became more obvious, leading to the local instability of the material in a narrow area. The head of the Wf/Zr-MG rod was continuously sharpened with the increasing penetration depth. This phenomenon decreased the penetration resistance, and the penetration efficiency improved. Thus, the erosion process of the Wf/Zr-MG rod at high speed impact was revealed.

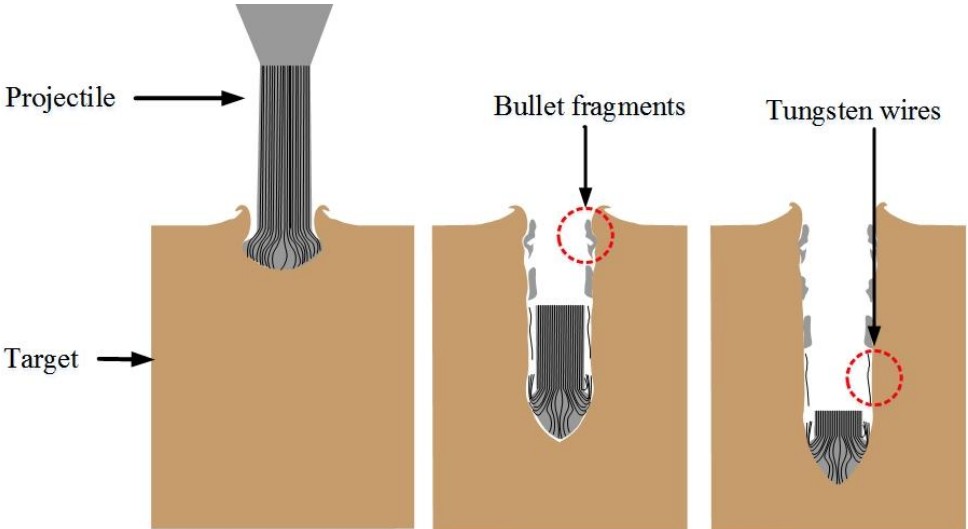

**Figure 12.** Penetration progress of the RHA by Wf/Zr-MG projectile.

*5.2. Theoretical Analysis of Penetration Resistance*

According to the conservation laws of energy, the rod meets the requirements in the process of penetration, as follows:

$$E = E_p + E_f, \tag{1}$$

where $E$ is the kinetic energy after the rod exits the muzzle. The above section shows that the kinetic energy of the two rod materials is approximately equal. $E_p$ is the energy used by the rod in the penetration process. $E_f$ is the energy consumed by the rod to resist the penetration resistance during penetration.

$$E = E_p - E_f \tag{2}$$

where $f(p)$ is the energy consumed by the rod to resist the axial resistance during the process of penetration. The $f(Fz)$ is the energy consumed by the rod to resist radial resistance during the process of penetration.

According to the previous test results and literature [32], the axial resistance of 93W rod in the process of penetration is as follows:

$$P_W = \frac{\pi D_W^2}{4}\left(AY_t + \frac{1}{2}B\rho_t u^2\right) \tag{3}$$

While the axial resistance of the Wf/Zr-MG rod in the process of penetration is as follows:

$$P_{Wf} = \frac{\pi D_{Wf}^2}{4}\left(AY_t + \frac{B\rho_t^{u^2} D_{Wf}^2}{4b^2 + D_{Wf}^2}\right) \tag{4}$$

where $P$ is the target plate resistance; $D$ is the crater diameter; $Y_t$ is the dynamic buckling strength of the target material; $\rho_t$ is the density of the target plate; $u$ is penetration velocity; $A$ and $B$ are the two constants of target resistance calculated by cavity expansion model; and $B$ is the head length of projectile residue.

$$g = P_{Wf} - P_W \tag{5}$$

By substituting the parameters in Table 2, $g < 0$, the penetration axial resistance of the Wf/Zr-MG rod became lower than that of the 93W rod during the process of penetration,

under the assumption that the action distance of axial resistance of the two rod materials are equal. The $f(P_w) > f(P_{wf})$. The angle mark w represents 93W, and Wf represents Wf/Zr-MG.

$$F_Z = \int_{Z=0}^{Z=l_k} [P_s + P_I(\eta)] \sin(\eta) \tag{6}$$

Among them, $p_s$ is the static resistance, which is only related to the material properties of the target plate. For linear hardening materials, their value is as follows:

$$P_S = \frac{2}{3}\sigma_{S,C}(1 + \ln\frac{2E}{3\sigma_{S,C}}) + \frac{2}{27}\pi^2 E_t \tag{7}$$

where $\sigma_{S,C}$ is the yield stress of the target plate, $E$ is the elastic modulus of the target plate, and $E_t$ is the hardening modulus of the target plate.

PI is the dynamic pressure, and its vector direction is perpendicular to a plane that passes through the center of the circle of the curve of the rod nose and with a η angle to the projectile diameter.

$$P_I = \rho_t(B_1 \frac{D_c x''}{2} + B_2 V_p^2) \tag{8}$$

where $x''$ is the negative acceleration of the rod, and $B_1$ and $B_2$ are the inertia coefficients. The $v_p$ is the particle velocity adjacent to the rod surface in the target.

The target materials used in the test are the same. Thus, the static resistance of the target is equal, and the influence parameters $\rho_t$ of dynamic pressure are equal. At the end of the process of the stage of launching, if it is assumed that the impact velocity of the 93W rod and Wf/Zr-MG rod are equal, then the $v_p$ is equal, and the action distance of each resistance is equal. For the incompressible materials, $B_1 = 1$ and $B_2 = 1.5$. It is assumed that the acceleration of two rod materials are the same at the end of the process of the stage of launching; the crater shape caused by the penetration of the two rod materials is also the same. Therefore, PI and Fz are the increasing functions related to the crater diameter $D_c$. So, when $D_w > D_{wf}$, $F_{zw}$ is also $> F_{zwf}$, $f(F_{zw}) > f(F_{zwf})$, $Ef_w > Ef_{wf}$, and $E_w > E_{wf}$.

In conclusion, the theoretical analysis shows that the nose shape, crater diameter, and penetration resistance of Wf/Zr-MG rods were less than those of the 93W rod. Therefore, under the same conditions, the penetration ability of the Wf/Zr-MG rod is better than that of the 93W rod.

## 6. Conclusions

The penetration ability and erosion mode of the 93W and Wf/Zr-MG rods were compared. The penetration experiments were carried out on two kinds of rods. According to the macroscopic and microscopic analysis results of the penetrator residual and the craters, we can obtain the following conclusions:

(1) Under the same impact velocity, the self-sharpening effect of the Wf/Zr-MG rod is obvious. The diameter of the crater penetrated by the Wf/Zr-MG rod is smaller, and the penetration resistance is less than that of the 93W rod under the same conditions. The penetration depth is approximately 10% higher than that of the 93W rod.

(2) Under the same impact velocity, the failure modes of the two rod materials are completely different. The penetration failure mode of the Wf/Zr-MG rods into RHA is the bending, backflow, and fracture of the WFs. The bottom of the crater penetrated by Wf/Zr-MG rods is conical. The failure mode of the 93W rod is the area with a large deformation on both sides of the "mushroom nose," which is constantly broken and peeled during the penetration process. The bottom of the crater is hemispherical.

(3) Under the same impact velocity, the extrusion ability of the two kinds of rod materials to the side wall of the crater are evidently different. The MRSL formed by 93W in the process of penetration shows a wavy change. It increases the backflow resistance of the rod and target materials and increases the pressure of the rod on the side wall of target plate and the penetration resistance. By contrast, the MRSL is more

relatively straight and smooth when formed by the Wf/Zr-MG rods, and the backflow of projectile and target materials is easy. The pressure of the rod on the side wall of the crater is lower, which reduces the penetration resistance and significantly improves the penetration ability.

**Author Contributions:** Conceptualization, F.Z., Z.D. and C.D.; Methodology, F.Z., Z.D. and C.D.; Investigation, F.Z., C.D., Z.D. and G.G.; Writing—original draft, F.Z., C.D., Z.D. and G.G.; Writing—review and editing, C.D. and Z.D.; Formal analysis, F.Z., C.D. and C.C.; Funding acquisition, C.D. and Z.D.; Project administration, C.D. and G.G.; Software, F.Z., C.D. and X.W.; Supervision, C.D., Z.D. and G.G.; Curation, F.Z., C.C. and X.W. All authors have read and agreed to the published version of the manuscript.

**Funding:** This research was funded by the National Natural Science Foundation of China, (Grant No.12102201, 11802130, 11772160).

**Institutional Review Board Statement:** This study does not involving humans or animals.

**Informed Consent Statement:** Not applicable.

**Data Availability Statement:** Not applicable.

**Acknowledgments:** We wish to express our gratitude to the members of our research team, Zhaojun Pang, Xi Chen, Shuai Yue, and Jiangbo Wang.

**Conflicts of Interest:** The authors declare no conflict of interest.

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
