# Peer review of "Penetration Gain Study of a Tungsten-Fiber/Zr-Based Metallic Glass Matrix Composite"

_crystals, doi:10.3390/cryst12020284_

Round 1

Reviewer 1 Report

Please change the title to Penetration Gain Study of Tungsten-fiber/Zr-Based Metallic Glass Matrix Composite as no mechanism is described

It is not novel study. Similar studies related to kinetic energy penetrator (KEP) were carried out earlier as well which authors have not cited.

Rafique, M.M.A. 2018. Bulk Metallic Glasses and Their Composites: Additive Manufacturing and Modeling and Simulation. Momentum Press, United States.

Kecskes, Laszlo & Edwards, Brian & Woodman, Robert. (2004). Hafnium-Based Bulk Metallic Glasses for Kinetic Energy Penetrators. 19. 10.1142/9789812772572_0033.

Dandliker, Richard B. (1998) Bulk metallic glass matrix composites : processing, microstructure, and application as a kinetic energy penetrator. Dissertation (Ph.D.), California Institute of Technology. doi:10.7907/HTJS-N846. https://resolver.caltech.edu/CaltechETD:etd-01242008-074925

Cai W. D.; Li Y.; Dowding R. J.; Mohamed F. A.; Lavernia E. J. (1995). "A review of tungsten-based alloys as kinetic energy penetrator materials". Rev. Particulate Mater3: 71–131.

Author Response

School of Mechanical Engineering

Nanjing University of Science and Technology

Nanjing, Jiangsu, 210094, P.R. China

January. 24th, 2022

Dear Dr. Daniele Massella

My co-workers and I greatly appreciate your e-mail informing us of the review results of our manuscript. Now, I am enclosing a new version of the manuscript revised in line with the reviewers’ comments, together with a list of those revisions.

Please feel free to let me know if you need us to address any further issues.

With best regards,

Dr. Chengxin Du

Authors: Feng Zhou, Chengxin Du, Zhonghua Du, Guangfa Gao, Chun Cheng, and Xiaodong Wang

Title: Penetration Gain Study of Tungsten-fiber/Zr-Based Metallic Glass Matrix Composite

Submitted to: Crystals

First of all, we would like to thank the reviewers for their professional reviews and insightful comments, which have led us to improve the overall quality of our paper. We have carefully revised the paper in line with those comments one by one as follows.

To reviewer 1

Comment 1:

Please change the title to Penetration Gain Study of Tungsten-fiber/Zr-Based Metallic Glass Matrix Composite as no mechanism is described.

Response:

Line 2, the title has been changed as “Penetration Gain Study of Tungsten-fiber/Zr-Based Metallic Glass Matrix Composite”.

Comment 2:

It is not novel study. Similar studies related to kinetic energy penetrator (KEP) were carried out earlier as well which authors have not cited.

  • Rafique, M.M.A. 2018. Bulk Metallic Glasses and Their Composites: Additive Manufacturing and Modeling and Simulation. Momentum Press, United States.
  • Kecskes, Laszlo & Edwards, Brian & Woodman, Robert. (2004). Hafnium-Based Bulk Metallic Glasses for Kinetic Energy Penetrators. 19. 10.1142/9789812772572_0033.
  • Dandliker, Richard B. (1998) Bulk metallic glass matrix composites : processing, microstructure, and application as a kinetic energy penetrator. Dissertation (Ph.D.), California Institute of Technology. doi:10.7907/HTJS-N846. https://resolver.caltech.edu/CaltechETD:etd-01242008-074925

()Cai W. D.; Li Y.; Dowding R. J.; Mohamed F. A.; Lavernia E. J. (1995). "A review of tungsten-based alloys as kinetic energy penetrator materials". Rev. Particulate Mater. 3: 71–131.

Response:

References (1) has been added on line 50 of the article as references 13.

References (2) has been added on line 50 of the article as references 14.

References (3) has been added on line 50 of the article as references 15.

References (4) has been added on line 41 of the article as references 8.

The line 68 “which has not been studied” is deleted, and add “which the results only have been listed in relatively few articles,such as literature [13-15]”.

Reviewer 2 Report

The article is devoted to the study of the properties of high-temperature composites based on Zr (Wf/Zr-MG). These studies are of scientific interest and practical significance, since the selected compositions have a high potential for industrial use. However, before the article is accepted for publication, the authors should make changes and answer the questions posed.
1 In the introduction, the authors should write in more detail the purpose and relevance of this work, taking into account recent achievements in this field.
2 Authors should provide additional images of the internal surface of the samples after damage in order to present the results of damage in more detail.
3 Authors should explain why there is such a heterogeneity in the distribution of elements, and should also present the results of mapping, which will reflect in more detail the results of the distribution of elements in the structure.
4 The authors should explain why the hardness does not change linearly, and a thin hardened layer of material is observed at the site of damage? What is the reason for its formation, does deformation processes in case of damage affect it?
5 The authors should give an explanation of what exactly is connected with the formation of projectile fragments when they hit a composite target, and also where do the tungsten fragments come from along the projectile trajectory?

Author Response

School of Mechanical Engineering

Nanjing University of Science and Technology

Nanjing, Jiangsu, 210094, P.R. China

January. 24th, 2022

Dear Dr. Daniele Massella

My co-workers and I greatly appreciate your e-mail informing us of the review results of our manuscript. Now, I am enclosing a new version of the manuscript revised in line with the reviewers’ comments, together with a list of those revisions.

Please feel free to let me know if you need us to address any further issues.

With best regards,

Dr. Chengxin Du

Authors: Feng Zhou, Chengxin Du, Zhonghua Du, Guangfa Gao, Chun Cheng, and Xiaodong Wang

Title: Penetration Gain Study of Tungsten-fiber/Zr-Based Metallic Glass Matrix Composite

Submitted to: Crystals

First of all, we would like to thank the reviewers for their professional reviews and insightful comments, which have led us to improve the overall quality of our paper. We have carefully revised the paper in line with those comments one by one as follows.

To reviewer 2

Comment 1:

In the introduction, the authors should write in more detail the purpose and relevance of this work, taking into account recent achievements in this field.

Response:

The text in “INTRODUCTION” The lines 68-73 is added.“ in order to enrich the research results in this direction, summarize and refine the previous research contents, and provide a new research idea for the research on the penetration mechanism of penetrators, it is particularly important to deeply explore the microstructure analysis of targets in the penetration process”

Comment 2:

Authors should provide additional images of the internal surface of the samples after damage in order to present the results of damage in more detail.

Response:

The text in “Experiment” The line 182 is added Internal surface morphology and EDS spectra of crater as .Fig. 7.

The text in “Experiment” The line 202 is added Internal surface morphology and EDS spectra of crater as Fig. 9.

Comment 3:

Authors should explain why there is such a heterogeneity in the distribution of elements, and should also present the results of mapping, which will reflect in more detail the results of the distribution of elements in the structure

Response:

The text in “Micro-Structure analyses on the crater” The lines 174-181 are added “Among of them, EDS 1 - 3 in Fig. 6 reflects the element distribution of bright inclusions in MSTL, MSTL and matrix materials respectively, and the EDS 1 in Fig. 7(b) is also reflect the element distribution of MSTL. The main composition in EDS 1 point was W, prove the bright part in the figure is broken tungsten particles condensed in MSTL. It can be seen from Fig. 6 (c) (d) that MSTL is similar to the matrix material, and its main component is Fe. Fig. 6 and Fig. 7 shows that the MRSL was composed of the melting target material and broken bulk tungsten particles.”

Comment 4:

The authors should explain why the hardness does not change linearly, and a thin hardened layer of material is observed at the site of damage? What is the reason for its formation, does deformation processes in case of damage affect it?

Response:

For question “The authors should explain why the hardness does not change linearly”. The text in the lines 221-223 are added “The pressure of WHA rod on the side wall of crater is not linearly, resulting in the hardness of the wall of the craters penetrated by WHA rod does not change linearly” 

For question “explain a thin hardened layer of material is observed at the site of damage”. The text in the lines 223-225 are added “It shows that the pressure of projectile and MSTL improve the hardness of crater surface.” in the lines 230-231 are added “the higher temperature make the hardness peak value of the side wall of the crater is reduced and the hardness decreases in a gradient”

Comment 5:

The authors should give an explanation of what exactly is connected with the formation of projectile fragments when they hit a composite target, and also where do the tungsten fragments come from along the projectile trajectory?

Response:

In the process of penetration, the compressive stress in the contact area between the projectile and target was more than the strength limit of the penetrator. The 93W rod showed good dynamic plasticity during the severe deformation. The nose shape significantly deformed, became blunter, flowed to the side, and became a clear “mushroom” shape. With increasing penetration depth, the severe plastic deformation continued to occur in the “mushroom” nose of the rod. The materials on the side of “mushroom” nose continued to erode and fell off from the rod. Tungsten fragments of different sizes are formed. The larger tungsten fragments are discharged from the crater under the reaction force of projectile impact, while the smaller tungsten fragments are coagulated on the surface of MSTL, and then the above process is repeated.

Reviewer 3 Report

In this work, Tungsten fiber/Zr-based bulk metallic glass matrix composite (Wf/Zr-MG) was used as an enhanced potential 12 penetrator material. This manuscript is written and presented very well. However some experimental needed to justify the microstructural features of Wf/Zr-MG composite. In section 2.3 (projectile and target material), X-ray diffraction (XRD) and Transmission electron microscopy (TEM) must be carried out to confirm the BMG composites. With some major modifications, this manuscript can be published in this esteemed "crystals" journal. 

Author Response

School of Mechanical Engineering

Nanjing University of Science and Technology

Nanjing, Jiangsu, 210094, P.R. China

January. 24th, 2022

Dear Dr. Daniele Massella

My co-workers and I greatly appreciate your e-mail informing us of the review results of our manuscript. Now, I am enclosing a new version of the manuscript revised in line with the reviewers’ comments, together with a list of those revisions.

Please feel free to let me know if you need us to address any further issues.

With best regards,

Dr. Chengxin Du

Authors: Feng Zhou, Chengxin Du, Zhonghua Du, Guangfa Gao, Chun Cheng, and Xiaodong Wang

Title: Penetration Gain Study of Tungsten-fiber/Zr-Based Metallic Glass Matrix Composite

Submitted to: Crystals

First of all, we would like to thank the reviewers for their professional reviews and insightful comments, which have led us to improve the overall quality of our paper. We have carefully revised the paper in line with those comments one by one as follows.

To reviewer 3

Comment 1:

In this work, Tungsten fiber/Zr-based bulk metallic glass matrix composite (Wf/Zr-MG) was used as an enhanced potential 12 penetrator material. This manuscript is written and presented very well. However some experimental needed to justify the microstructural features of Wf/Zr-MG composite. In section 2.3 (projectile and target material), X-ray diffraction (XRD) and Transmission electron microscopy (TEM) must be carried out to confirm the BMG composites.

Response:

The text in “Experiment” The line 112 is added X-ray of the composite (c) and TEM electron diffraction pattern of Zr-MG (d). and the lines 103-109 is added a sentence. “Fig. 3 (c) shows the X-ray diffraction pattern of the composite. The body centered cubic (bcc) diffraction peaks for the Wf are superimposed on the broad diffuse scattering hump, which is typical characteristic from the amorphous structure. The BMG cross-sectional high resolution TEM pattern are shown in Fig. 3 (d). Broad and diffuse diffraction rings are clearly seen in the TEM patterns implying its amorphous structure, which is consistent with the XRD analysis results. ” to explain the pictures.

Round 2

Reviewer 1 Report

.

Reviewer 2 Report

The authors answered all the questions posed. The article may be accepted for publication.

Reviewer 3 Report

In this work, Tungsten fiber/Zr-based bulk metallic glass matrix composite (Wf/Zr-MG) was used as an enhanced potential 12 penetrator material. This manuscript is written and presented very well. Now this research article may be accepted in the present form.